# Intrinsic Blue Fluorescence of 2.0G PAMAM-DCM Polymer Dots and Its Applications for Fe^3+^ Sensing

**DOI:** 10.3390/s22031075

**Published:** 2022-01-29

**Authors:** Xin Wang, Weiguang Shi, Yuda Wang, Dan Cheng, Jiahui Liu, Shihan Xu, Wei Liu, Biao Dong, Jiao Sun

**Affiliations:** 1College of Chemistry & Chemical Engineering, Northeast Petroleum University, Daqing 163318, China; wx18756130799@163.com (X.W.); L19845920393@163.com (J.L.); 2Key Laboratory of Continental Shale Hydrocarbon Accumulation and Efficient Development, Ministry of Education, Northeast Petroleum University, Daqing 163318, China; 3Department of Cell Biology, College of Basic Medical Sciences, Jilin University, Changchun 130021, China; wangyuda2022@163.com; 4Daqing Ecological Environment Monitoring Center, Daqing 163318, China; chengdan83@163.com; 5Department of Bioengineering, University of Washington, Seattle, WA 98195, USA; xushihan@uw.edu; 6State Key Laboratory on Integrated Optoelectronics, College of Electronic Science and Engineering, Jilin University, Changchun 130012, China; liu_wei21@mails.jlu.edu.cn (W.L.); dongb@jlu.edu.cn (B.D.)

**Keywords:** PAMAM, polymer dots, fluorescent mechanism, metal ionic sensing

## Abstract

A typical and environment-friendly fluorescent polyamine-amine (PAMAM) features good compatibility and unique surface modification, while it is restricted by a low fluorescence property performance and an unclear fluorescence mechanism. In this work, we prepared blue fluorescent PAMAM polymer dots (PDs) via a simple hydrothermal method based on dichloromethane (DCM) and 2.0G PAMAM. The quantum yield achieved was 32.1%, which was 25 times stronger than that of 2.0G PAMAM due to the lone-pair electron leap of the amine groups, the aggregation of carbonyl groups, as well as the crosslinking induced by DCM inside the PAMAM. In addition, the fluorescent 2.0G PAMAM-DCM PDs show a great Fe^3+^ sensing property with the detection limit of 56.6 nM, which is much lower than the safety limits (5.36 μM) in drinking water, indicating its great potential for Fe^3+^ detection in aqueous media.

## 1. Introduction

Polyamide-amine (PAMAM) is a highly symmetrical macromolecule with an internal cavity: exhibits good compatibility, unique hydrodynamic properties, and easy modification properties [1,2,3]; receives increasing attention in bioimaging [4,5], sensors [6], catalysis [7,8], drug-delivery [9,10], etc. Previously, Wang and Imae [11] reported that PAMAM, with different terminal groups, including −NH_2_, −C=O, or −OH, could present weak and blue fluorescence. PAMAM-based fluorescent carbon/polymer materials can be prepared by the surface modification method to obtain efficient fluorescence, suggesting that the fluorescence properties are related to the generations of PAMAM and pH [12,13,14]. These studies are helpful to gain a better understanding of the relationship between the internal backbone and fluorescence. Recent studies have concluded that the fluorescence properties of PAMAM will be affected by special conditions, including the oxidation of tertiary amines, photoelectron transfer, and the aggregation of internal carbonyl groups [15,16,17]. This indicates that at least two fluorescence sites are inside the PAMAM, influenced by pH or temperature [18,19,20,21]. Therefore, the stability of PAMAM and its fluorescent property will be limited.

The development of polymer carbon dots (PCDs, PDs) solved the unstable fluorescence problem of polymers [22,23]. Note that polymer carbon dots are zero-dimensional and mildly carbonized nanomaterials with unique core–shell structures composing internal carbon and external polymer, which can greatly enhance the fluorescence property and stability [24,25]. With external shell modification, carbon dots can be further multi-functionalized.

Meanwhile, PCDs, as a fluorescence sensor, [26,27] show great applications in environmental sciences [28,29], chemical analysis [30], and biological fields [31,32,33]. In contrast to the instrumental detection of metal ions, PD fluorescence sensors are simple, portable, highly efficient, and fast-responding [34]. Most organics-based fluorescent probes are hydrophobic and toxic, resulting in poor water solubility and low detection sensitivity [35]. PAMAM-based PDs could be a good candidate as a fluorescent sensor with improved detection performance for metal ions, owing to their good compatibility, nontoxicity, and easy surface modification. However, for better sensing performance, the fluorescence efficiency of PAMAM should be improved. Moreover, the utilization of PDs is still hindered by unclear fluorescent mechanisms due to their complicated structures, which is also an urgent problem that needs to be solved.

In this paper, dichloromethane (DCM) was employed as raw material to react with the amino group of PAMAM via a nucleophilic substitution reaction. Diaz et al. reported that DCM is easier to react with a tertiary amine, followed by a secondary amine, which is difficult to react with a primary amine [1]. Thus, DCM is a suitable candidate for investigating the internal fluorescence mechanism of PAMAM. In order to enhance the fluorescence intensity of PAMAM-based PDs, 2.0G PAMAM and DCM were combined to obtain 2.0G PAMAM-DCM PDs by the one-pot hydrothermal method. It exhibits remarkable water solubility, strong blue fluorescence, and pH- and temperature-responsive properties. The intrinsic fluorescence mechanism of 2.0G PAMAM-DCM PDs was investigated by focusing on the amine group and carbonyl group. DCM can restrict the transfer, rotation, and vibration of lone-pair of electrons on the amine group, promoting the crosslinking and aggregation of functional groups in PAMAM and presenting efficient fluorescent properties (quantum yield of 32.1%). Moreover, the fluorescent properties of 2.0G PAMAM-DCM PDs were studied under different conditions, such as pH, temperature, and concentrations. In addition, the enhanced sensing of Fe^3+^ ions was achieved with a detection limit of 56.6 nM, which indicates its potential applications in environmental monitoring.

## 2. Materials and Methods

### 2.1. Reagents

Methyl acrylate (98%), ethylenediamine (99%), methanol (98%), dichloromethane (99%), hydrochloric acid (AR, 35 wt%), sodium hydroxide (98%), H_3_PO_4_ (AR), KCl (AR), HgCl_2_ (AR), CaCl_2_(AR), NaCl (AR), CuCl_2_(AR), MgCl_2_ (AR), CdCl_2_ (AR), HgCl_2_ (AR), ZnCl_2_ (AR), FeCl_3_, and AlCl_3_ (AR) were purchased from Aladdin Reagent Co., Ltd. (Shanghai, China). Disodium hydrogen phosphate (98%), sodium dihydrogen phosphate (98%), and quinine sulfate (98%) was purchased from Macklin Biochemical Co., Ltd. (Shanghai, China). Deionized water was made in our laboratory.

### 2.2. Experimental Methods

#### 2.2.1. Synthesis of 0.5G PAMAM, 1.0G PAMAM and 2.0G PAMAM

Different generations of PAMAM were synthesized by a divergent method. The synthesis sequence of PAMAM products is 0.5G PAMAM→1.0G PAMAM→2.0G PAMAM. Firstly, a Michael addition reaction was performed. Ethylenediamine was dissolved with 1.2 g into 50 mL methanol solution. Then, they were transferred to a cold-water bath in a three-necked flask for 30 min. Methyl acrylate (13.5 g) was added dropwise through a constant pressure funnel and stirred at 30 °C for 36 h. The excess methyl acrylate and methanol were evaporated by the reduced pressure distillation method at 55 °C, and 7.8 g 0.5G PAMAM products were obtained. Secondly, 4.04 g of 0.5G PAMAM was dissolved in 60 mL methanol solution; 5.16 g of ethylenediamine was added dropwise to the above solution and stirred for 30 min. The reaction was carried out at 30 °C for 24 h. The solution was evaporated by rotation at 75 °C, and 5.0 g of 1.0G PAMAM was obtained. Finally, 2.0G PAMAM was synthesized through the repeated steps above.

#### 2.2.2. Synthesis of 2.0G PAMAM-DCM PDs

The 2.0G PAMAM-DCM PDs were obtained via a hydrothermal method. 2.0G PAMAM-NH_2_ (0.2 mM) were dissolved for use. Then, PAMAM (aq) and 5.0 mL dichloromethane were added into a stainless-steel autoclave (25 mL) for oil bath; the pH of the solutions was adjusted using HCl solution (0.1 M) or NaOH (0.1 M). The resulting solution was then treated by a one-step hydrothermal method under different conditions, including temperature (60–140 °C), pH (3–11) and concentrations (0.01–0.1 mM). The pure PDs were obtained by centrifugation and vacuum rotary evaporation, and freeze-dried.

### 2.3. Measurements

The fluorescence spectra of the resultant polymers were recorded at room temperature on a fluorescence spectrophotometer (LS-55, Perkin–Elmer, Wellesley, MA, USA). The slits of excitation and emission were 10.0 nm and 2.5 nm, respectively, with a scanning rate of 200 nm·min^−1^. All measurements were performed at room temperature. The morphology and the structure of the sample were analyzed by a high-resolution transmission electron microscope (HR-TEM) (Tecnai G^2^ F20 S-Twin, FEI, Hillsboro, FL, USA) at an accelerating voltage of 200 kV. UV–Vis absorption spectra were recorded using a UV spectrophotometer (UV–Vis) (UV-1700 PharmaSpec, Kemeijia, China). Ultra-pure water was used as the reference solution, and the wavelength range was 200–600 nm. ^1^H NMR spectra were collected on a 400-MHz Bruker NMR spectrometer (AV400, Bruker, Ettlingen, Germany), with the freeze-dried product dissolved in D_2_O. Fourier transform infrared spectra (FT-IR) were performed on a Bruker FT-IR spectrometer (Tensor27, Bruker, Leipzig, Germany), and the scanning time was set to 5 times per minute. The pH values of the solutions were measured with a pH meter (PHS-3E, Shanghai Leici Instrument Company, Shanghai, China).

Quinine sulfate (0.1 M H_2_SO_4_ as the solvent, QY = 54%) was chosen as the standard. The QY of PDs (in water) was calculated by the reference point method:*ϕ*_x_ = (A_n_/A_x_)(∫*F*_x_/∫*F*_n_)*ϕ*_n_
where *ϕ* is the QY, A is the absorbance (0.01 < A < 0.05, to minimize self-absorption effects), and ∫*F* is the integrated area of the emission spectrum (λ_EX_ = 360 nm). The subscript “n” refers to the standard with known QY and “x” for the sample (PDs).

## 3. Results and Discussion

### 3.1. 2.0G PAMAM-DCM PDs

The 2.0G PAMAM-DCM PDs were prepared by the hydrothermal method at 100 °C. As shown in Figure 1A, the 2.0G PAMAM-DCM solution exhibits a stronger blue fluorescence under the UV light (λex = 365 nm), and the HR-TEM image shows a mild carbonization nano dot (2.12 nm) with no obvious crystal lattice, indicating that 2.0G PAMAM-DCM PDs are a kind of PDs, but not pure CDs. To further prove that the PDs retain the properties of the polymer, FTIR spectra and ^1^H NMR were analyzed for their molecular structures and functional groups. The absorption peak at 3284 cm^−1^ and 3080 cm^−1^ were attributed to the stretching vibration of –NH_2_, presenting in both 2.0G PAMAM-DCM PDs and 2.0G PAMAM. It indicated that the peripheral primary amines in PAMAM did not react with DCM. The stretching vibration of C=O (1643 cm^−1^) and C–N (1385 cm^−1^) co-existed in the two systems, suggesting that the structural characteristics of the polymer are preserved. Meanwhile, the peak near 2100 cm^−1^ was a newborn vibration from the reaction between tertiary amine and DCM [36]. The secondary amine could also bind with the DCM to form C–N (Figure 1B). Consequently, DCM promoted the crosslinking of intramolecular amine groups, the aggregation of carbonyl groups, and the formation of large bond conjugation, which reduced the electron cloud density in the systems [37]. Figure 1C documents a unified low-frequency shift of the peaks in the range of 2–4 ppm, and two new-born chemical shifts of 4.2 and 5.2 ppm, which resulted from the decreased electron density by polymer crosslinking, induced by the reaction between the tertiary amine or secondary amine in PAMAM and DCM. These observations suggest that 2.0G PAMAM-DCM are PDs with special features of the combination of carbon dots and polymers.

To further investigate the fluorescence properties and the fluorescence mechanism of 2.0G PAMAM-DCM PDs, experiments were carried out in different conditions, such as concentration, temperature, and pH.

### 3.2. Fluorescence Properties of 2.0G PAMAM-DCM PDs

The UV–Vis absorption spectra and the fluorescence spectra of 2.0G PAMAM and 2.0G PAMAM-DCM PDs are shown in Figure 2. It can be seen that two absorption bands in the 2.0G PAMAM-DCM PDs are consistent with: π→π* electronic transition (225 nm), as large as the π-conjugated off-domain system. It was formed by a combination of the crosslinking of intramolecular amine groups and the aggregation of carbonyl groups: the n→π* electronic transition (303 nm) corresponding to C=O. Additionally, there is only one absorption band which is assigned to the n→π* electronic transition of C=O in 2.0G PAMAM (Figure 2A). Meanwhile, the emission peak was significantly red-shifted by varying the excitation wavelengths from 240 to 410 nm (Figure 2B). When excited at 390 nm, it emitted blue light, with the emission peak located at 450 nm. When the emission wavelength was fixed at 450 nm, the excitation spectra showed large different intensities at two peaks, located at 260 nm and 390 nm (Figure 2C). When excited at 390 nm, weak fluorescence intensity was observed from 2.0G PAMAM, while the 2.0G PAMAM-DCM PDs solution appeared light-yellow in color and exhibited a strong blue fluorescence (Figure 2D).

#### 3.2.1. Effects of Concentrations on the Fluorescence Properties

According to previous research, the concentration of the precursor an important parameter that can affect the numbers and dispersion degree of fluorescent groups during the formation of PDs, resulting in a great fluorescent diversity [38]. Five different concentrations, with the range from 0.01 to 0.1 mol/L of 2.0G PAMAM, were employed for thermal reaction at 100 °C for 3 h. In Figure 3A,B, a progressive enhancement trend of fluorescence intensity was recorded with the increasing concentrations of 2.0G PAMAM, from 0.01 to 0.1 mol/L, and the yellow and blue color of the PDs solutions gradually deepened under sunlight and UV light, respectively. These observations suggest that the degree of the crosslinking of intramolecular amine groups and the aggregation of carbonyl groups increases with increasing concentrations of 2.0G PAMAM. To further verify the effect of self-quenching on fluorescence of the PDs solutions, the concentration of 2.0G PAMAM precursor was fixed at 0.1 mol/L; 2.0G PAMAM-DCM PDs were distributed into the water, obtaining the concentrations from 0.1 to 5 mg/mL. As shown in Figure 3C,D, the fluorescence intensity presented a curvilinear upward trend from 0.1 to 5 mg/mL. When the concentration was greater than 4 mg/mL, the fluorescence intensity decreased due to its self-quenching. These results are generally in agreement with the earlier research [39].

#### 3.2.2. Effects of Temperature on the Fluorescence Properties

Previous reports conclude that temperature is considered an important parameter to control the carbonization degrees. To investigate the influence of different reaction temperatures on the fluorescence properties of 2.0G PAMAM-DCM PDs, experiments were carried out at 60 °C, 80 °C, 100 °C, 120 °C, and 140 °C, respectively. In this work, 100 °C was the initial temperature to form 2.0G PAMAM-DCM PDs with the highest fluorescent intensity. The fluorescence intensities decreased at 120 °C and 140 °C. The solutions were scarcely colored under sunlight at 60 °C and 80 °C, with no fluorescent intensities (Figure 4A,B).

However, 2.0G PAMAM-DCM PDs began to transform into CDs when the reaction temperature was higher than 100 °C. When increasing the temperature from 120 °C to 140 °C, the average particle size of CDs regularly increased from 2.48 nm to 4.36 nm. Moreover, well-resolved lattice fringes were observed clearly in the HRTEM image (Figure 4C,D). The interplanar spacing also increased, from 0.18 nm to 0.23 nm, which is close to the (100) diffraction facets of graphite carbon. These processes of polymer carbonization are generally in agreement with earlier research [22,23]. The fluorescence intensities of 2.0G PAMAM-DCM PDs were different, but the emission positions did not change at 100 °C or 140 °C, suggesting the temperature-responsive fluorescence property. The results indicate that 100 °C is the optimal temperature for the stable molecular structure of 2.0G PAMAM-DCM PDs, while 120–140 °C is the optimal temperature for the stable fluorescence property of 2.0G PAMAM-DCM CDs.

#### 3.2.3. Effects of pH on the Fluorescence Properties

Inarguably, the crosslinking degree, the strength of the hydrogen bond, and the chemical reaction of functional groups of PAMAM molecules can be controlled by pH [11], by changing charge distributions, and by intrinsic emissions. Thus, pH is an emission-related factor that can affect fluorescence properties. In this case, the pH sensitivity of 2.0G PAMAM-DCM PDs was recorded, with the fluorescence spectrum at a fixed excitation wavelength of 390 nm. Figure 5A,C document a first upward and then downward tendency of fluorescence intensity in the pH range from 3.02 to 10.75. There are no fluorescence spectra shifts, and the maximum emission intensity is obtained at pH 7.24. Moreover, the emission intensity is high enough at pHs ranging from 3.02 to 10.75 and maintains stability and fluorescence intensity, which can extend to a wide range of applications, such as environmental monitoring. From the aspect of pH sensitivity, the amine groups are protonated in the condition of pH < 7; isomerization phenomena would happen to destroy the nucleophilicity of DCM [18]. In the condition of pH > 7, the fluorescence intensity becomes weak, owing to DCM restricting the electron movement of the amine group, as well as the rotation and vibration in PAMAM, weakening the crosslinking of the internal carbonyl group [40]. Meanwhile, compared with lower fluorescence intensities in the control system of 2.0G PAMAM (Figure 5B), this indicates that the reaction between DCM and amine groups can form large bond conjugations and crosslinking, which can enhance the fluorescence properties. In contrast to 1.0G PAMAM-DCM PDs, the results suggest that a similar variation tendency of fluorescence presents under the same pH, but the fluorescence intensity (Figure 5D–F), indicates that 2.0 G PAMAM has multiple amine and carbonyl functional groups to form fluorescent sites. These observations indicate that DCM reacts with the amines that are inside the PAMAM, but not the external −NH_2_.

### 3.3. Possible Fluorescence Mechanism of 2.0G PAMAM-DCM PDs

To identify if an important fluorescence center in 2.0G PAMAM-DCM PDs is induced by C=O bonds and their aggregations, a reaction (Figure 1) was conducted by reducing the C=O bond to an alcoholic hydroxyl group by sodium borohydride (NaBH_4_) to explore the variation of fluorescence properties.

An experiment was carried out by adding an excessive amount of NaBH4 in the 2.0G PAMAM-DCM PDs solution for 4 h at 40 °C. The different results before and after reductions are summarized in Figure 6A,B. A great decreasing tendency of fluorescence intensity was recorded after reduction, and the color of the PDs solutions gradually became colorless and weakly blue under sunlight and UV light, respectively. This indicated that a massive amount of fluorescence centers, caused by n→π * transitions, had vanished, together with a part of large π-bond conjugation induced by DCM, which is proved by the UV spectrum. As a result, the UV adsorption bands at 225 nm and 303 nm disappeared, which was attributed to π→π* electronic transition. The n→π* electronic transition also indicated that the π-conjugated off-domain system was destroyed, C=O was reduced, and the combination of the crosslinking of intramolecular amine groups and the aggregation of carbonyl groups was dissolved (Figure 6C). Only the crosslinking of intramolecular amine groups caused by DCM was residually viable at the 278-nm adsorption band, emitting the blue light.

Consequently, the proposed reaction mechanism is summarized in Figure 6D. It is clear that there are two fluorescent centers and a crosslink-enhanced emission in 2.0G PAMAM-DCM PDs. One is the C=O bonds, which emit a light-yellow fluorescence. Another is the π-conjugated off-domain systems obtained by the reaction among DCM, C=O, and internal amine groups of PAMAM, generating a blue light. The other is the crosslink-enhanced emission combined with the reaction of DCM and internal multiple amines, and the aggregation of carbonyl groups, enhancing the fluorescence. Combinations of multiple factors make the 2.0G PAMAM-DCM PDs exhibit a stronger blue fluorescence.

As a result, DCM is the key role in the fluorescence emission of 2.0G PAMAM-DCM PDs. The internal factors that dominated the emission may result from DCM: (1) the reaction of DCM and internal amine groups (−NH, −NR_2_) of PAMAM, but not the external primary amine (−NH_2_), which is controlled by the nucleophilic property of DCM; (2) DCM induces the crosslinking due to the shortened distances of branched chains of PAMAM; (3) DCM restricts the transfer, rotation, and vibration of lone-pair electrons on the amine group, causing the aggregation of carbonyl groups.

### 3.4. Metal Ion Sensing Performance

Owing to a good water solubility and fluorescent property of 2.0G PAMAM-DCM PDs, the chemical-sensing performance of the 2.0G PAMAM-DCM PDs was tested with different metal cations, including alkali metal ions (Na^+^ and K^+^), alkaline earth metal ions (Ca^2+^ and Mg^2+^), and transition metal ions (bivalence: Cu^2+^, Cd^2+^, Zn^2+^, Hg^2+^; trivalence: Fe^3+^ and Al^3+^). Figure 7A,B show that the fluorescence intensities of Hg^2+^, K^+^, Mg^2+^, and Cu^2+^ slightly decreased, and Fe^3+^ greatly quenched the fluorescence, suggesting that Fe^3+^ ions have a selective quenching effect on 2.0G PAMAM-DCM PDs. The photographs captured under UV light illustrated a strong blue color that was quenched to weak blue, which was generally in agreement with the phenomenon of NaBH4 reduction. This indicates that carbonyl groups could intimately bind with Fe^3+^ ions, and destroy the aggregation and the n→π* electronic transition of the C=O bond (yellow fluorescence center). Thus, 2.0G PAMAM-DCM PDs can specifically recognize Fe^3+^ ions and form stable complexes.

In addition, the high selectivity of the fluorescence probe depends on the precise limit of detection. The detection limits of 2.0G PAMAM-DCM PDs to Fe^3+^ ions in water were recorded in Figure 7C. The intensity of blue light keeps decreasing until completely quenching, along with the increasing concentration of Fe^3+^ ions in the range of 10 μM to 300 μM. A linear tendency between the Fe^3+^ concentration and fluorescence intensity of 2.0G PAMAM-DCM PDs presented. A linear equation describing the relationship between fluorescence intensity and Fe^3+^ concentration was obtained by fitting the data. Y = −0.69434X + 0.01478, R_1_^2^ = 0.9968, where Y is fluorescence intensity of 2.0G PAMAM-DCM PDs, X (μM) is Fe^3+^ concentration. Based on this tendency, a corresponding linear Stern–Volmer equation (F-F_0_)/F = 0.00199X + 0.02339, R_2_^2^ = 0.9976 can be designed to obtain the detection limit. Where X (μM) is Fe^3+^ concentration, F and F_0_ represent the fluorescence intensity of 2.0G PAMAM-DCM PDs in the absence and presence of Fe^3+^, respectively. The Stern–Volmer constant (s) is 0.00199. As a result, the detection limit is 56.6 nM (nM level), which represents the promotion of two orders of magnitude of the minimum value of Fe^3+^ (5.37 μM) in drinking water released by the US Environmental Protection Agency. Consequently, the 2.0G PAMAM-DCM PDs probes can extend biological applications and environmental monitoring.

## 4. Conclusions

In this work, 2.0G PAMAM was employed as a carbon source to improve the compatibility and modification of PDs, and DCM was used to enhance the fluorescence properties, to obtain blue 2.0G PAM-DCM PDs by a one-step hydrothermal method. The quantum yield of 2.0G PAMAM-DCM PDs was 32.1%, which is greatly enhanced—25 times that of the 2.0G PAMAM—also exhibiting remarkable pH- and temperature-responsive fluorescence properties. DCM played a key role in limiting the unshared electron transition of the internal amine functional groups as well as their rotations and vibrations. It formed the strongest and most stable fluorescence intensity under neutral conditions at 100 °C. At the same time, the PDs can be transformed into carbon dots higher than 100 °C, indicating that the carbonization degree would be accelerated from 2.0G PAMAM-DCM molecules on the shell. In addition, 2.0G PAMAM-DCM PDs show remarkable Fe^3+^ sensing properties in aqueous media, with great selectivity and the LOD of 56.6 nM, which is much lower than the international minimum standard (μM level). The design strategy of 2.0G PAMAM-DCM is significant for controlling structural modification, improving fluorescence properties of PDs, and extending applications for biological tracing and environmental assessment.

## Data Availability

Data can be made available upon request from the corresponding author.

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
