# Peer review of "Intrinsic Blue Fluorescence of 2.0G PAMAM-DCM Polymer Dots and Its Applications for Fe3+ Sensing"

_sensors, 2022, doi:10.3390/s22031075_

Round 1
Reviewer 1 Report
The authors report blue-and-green photoluminescence from polyamine-amine (PAMAM)treated with dichloromethane as a cross-linker that can specifically detect Fe3+ ion among ten kinds of metal ions, in which blue photoluminescence decreases as Fe3+ increases obeying apparent Stern-Volmer scheme. The work may involve an interesting topic, but might need a further clarification and additional experiments and careful analysis, and a great modification of main text/references/plots with captions in addition to English/grammatical corrections. I could not follow experimental section because of lack of the details and unkind description. Some readers cannot fully reproduce the current works and results. Because of scholarly incorrect wording, ambiguity, and poorer English, the present paper at the current stage would not meet a scientific high standard that is required from Sensors (MDPI). My comments are as follows.
1. First of all, For clarity/readability, should provide detailed chemical structures (not cartoons, Fig.6) of 1G/2G PANAM w/wo DCM used in this work.
2. Should give an explanation of role(s) of dichloromethane (DCM), not solvent. Did you check 1,2-dichloroethane DCE for comparison?
3. line 81 Ethylenediamine → ethylenediamine
4. line 92 ”1.2 g” of ethylenediamine was dissolved into 50 ml methanol solution
5. should avoid numerical characters in the beginning of sentence. Alternatively,
To 50 ml of methanol, 1.2 g of ethylenediamine was added.
I found similar numerical characters in the beginning of sentence and should modified.
6. line 95、excess methyl acrylate 95 and methanol were evaporated by rotary evaporation at 55°C
Do you mean ambient pressure or reduced pressure ?
7. line 109, “spin” centrifugation is strange. Should correct.
8. line 112, Please clarify location and country of Perkin-Elmer
9. line 126, Please clarify wavelength of UV light source.
10 line 150, did you measure PL excitation spectra monitored at several PL wavelengths ? For example, monitor wavelengths are chosen at 440, 480, and 500 nm. This is very important as to which electronic bands/defects of PANAM with/without DCM are responsible for ~400 nm and ~450 nm PL bands.
11. line 152, Please provide monitor wavelength for fluorescence excitation spectra.
12. line 175 Cure fitting ? Do you mean curve fitting ?
13. The authors use three technical terms, fluorescence, emission, and luminescence. Should unify wording.
14. line 327 line relationship ? should be linear relationship
15. Fig4D Possibly, Stern-Volmer plot. What does Wavelength (nm) in abscissa mean ? Concentration of Fe3+ ?
16. Should provide Stern-Volmer constant(s) for several ions including Fe3+.
17. In references, journals (full name) should be abbreviations that obey style format of MDPI.
18. All nine authors, at least two corresponding authors and the leading author should disclose ORCID along with his/her affiliation, education, and several representative publications. If not, should register/obtain his/her own ORCID to ensure his/her unique researchers’s names, affiliations, and research activities.
Reviewer 2 Report
The manuscript submitted by Wang et al. reports on the 2.0 generation Polyamidoamine (PAMAM) dendrimers that exhibit fluorescence properties upon complexation with dichloromethane molecules. Authors particularly highlight the fluorescence quenching effect induced by the Fe3+ ions that particularly qualifies the DCM-PAMAM complexes as effective sensors in detecting large amounts of Fe3+.
The Authors presents detailed experimental results that clearly demonstrate the molecular, as well as the physical properties of the PAMAM-DCM complexes under various temperature and pH conditions. Notably, Authors' hypothesis on the molecular origin of the fluorescence emission properties of the PAMAM-DCM complexes is experimentally supported. I particularly appreciate the carbonyl group reduction experiments carried out to explain the role of carbonyl groups in fluorescence emission mechanism; as well as the demonstration of the temperature dependent transition of the PAMAM-DCM polymer dots into carbon dots.
My only concern is the statement made in line 214, where the authors mention "well-resolved lattice fringes" of the carbon dots. I really can not see any fringes in the TEM images provided in Figure 4C. I would only suggest Authors to provide high-resolution TEM images to support their statement or somehow highlight the fringes that they saw in Figure 4C (maybe with an inset image).
I find the manuscript eligible for publishing in Sensors. However, a minor revision to remove the grammatical mistakes and typos would be required.
Reviewer 3 Report
This is an intriguing paper on the fluorescence properties of DCM-treated PAMAM (“polymer dots” according to the authors) and their application for Fe2+ sensing. Although almost contents of this paper are comprehensive, I did not understand well why they can conclude that the PDs can be transformed into carbon dots (l.338). Please show the appropriate data for this with clearly explaining the difference between the dendrimer (PAMAM) and the carbon dots.
Minor remarks
l.90 diverge -> divergent
l.103 DKG -> DCM
Fig.6 (A) and (B) in the figure Reducationn -> Reduced or Reduction (see also Fig.6 (C))
Round 2
Reviewer 1 Report
The original manuscript was properly revised in line with anonymous reviewers. The revision is likely to reach a publication level though several minor corrections may remain yet. The issue will be cleared at the editorial level.